# Perceived Barriers and Facilitators to Implementation of Inclusive Policy, Systems, and Environmental Changes

Cassandra Herman [1], Kerri Vanderbom [2], Karma Edwards [3] and Yochai Eisenberg [1,*]

1   Department of Disability and Human Development, University of Illinois at Chicago, 1640 W. Roosevelt Rd., Chicago, IL 60608, USA; cherma5@uic.edu
2   Prevention and Health Promotion, Butte County Public Health Department, Chico, CA 95928, USA; kvanderbom@buttecounty.net
3   Center for Advancing Health Communities, National Association of Chronic Disease Directors, Decatur, GA 30030, USA; kedwards_ic@chronicdisease.org
*   Correspondence: yeisen2@uic.edu

**Abstract:** People with disabilities (PWDs) are often excluded from health-promoting activities in their communities. Inclusive policy, systems, and environmental (PSE) changes can promote access to healthy lifestyle choices for PWDs. However, implementation of inclusive PSEs in community-based settings is challenging and we lack an understanding of what factors impact implementation of inclusive PSEs. The purpose of this study was to examine barriers and facilitators experienced by community coaches while planning and implementing inclusive PSEs. Semi-structured interviews (n = 10) were conducted with coaches as part of the Reaching People with Disabilities through Healthy Communities project. Interviews were coded using directed content analysis guided by the Theoretical Domains Framework and were categorized into barriers and facilitators within the COM-B framework (which identifies Capability, Opportunity, and Motivations as components that can impact Behavior). The opportunities domain, consisting of social influences and environmental context and resources, most impacted disability-inclusive PSE implementation. Within this domain, facilitators included community support, strong partnerships, technical assistance from experts, and alignment with ongoing initiatives. Barriers included the community's lack of knowledge about disability, fear regarding resources needed for inclusive changes, and lack of resources (time, staff, funding). Supports addressing the opportunities domain should be considered to facilitate the implementation of disability-inclusive PSEs to build healthy, accessible communities for all.

**Keywords:** disability; inclusion; health promotion; implementation

## 1. Introduction

One in four adults in the United States (US) report having a disability [1]. People with disabilities (PWDs) experience disparities in health outcomes compared to those without disabilities including reporting poorer health and higher rates of secondary conditions such as obesity and diabetes [2,3]. Unfortunately, there are fewer opportunities for PWDs to engage in health-promoting activities that can mitigate or prevent primary and secondary health conditions [4]. A variety of barriers make health-promoting activities inaccessible, including barriers at the community level and within the environment [5]. Environmental barriers such as lack of physical access (e.g., curb cuts, stairs), lack of knowledge and awareness of the needs of PWDs, and stigma related to disability [6,7] can lead to unwelcoming physical and social environments. While the environment is known to affect health by either deterring or promoting healthy behaviors [8], environmental barriers limit opportunities for PWDs to engage in health-promoting activities.

Public health initiatives have attempted to address some environmental barriers, through larger policy, systems, and environmental changes (PSEs), a health promotion approach with the potential to have a greater reach than individual-level interventions [9].

However, people with disabilities still consistently experience barriers within their communities when their needs are not considered throughout the process of planning and implementing PSEs. Centering the needs of PWDs in PSE implementation is needed to promote inclusion and accessibility. Recently, the Reaching People with Disabilities through Healthy Communities project (DHC) [10], a project of the National Association of Chronic Disease Directors (NACDD), funded ten US communities to intentionally include PWDs throughout PSE planning and implementation to increase disability inclusion and access to health-promoting activities. Each community aimed to implement three PSE changes guided by a multiphase implementation process. Within each community, two local community coaches, one from public health and one from a disability-serving organization, facilitated the planning and implementation process alongside existing multisectoral and cross-disciplinary community coalitions. Using the structured process, the coaches and coalitions of each community worked together to identify inclusion and access related needs, plan PSE changes to address those needs, develop action plans for implementing the solutions and implement the inclusive PSE changes. Community coaches also received technical assistance from their state's Disability and Health Program [11,12], NACDD, and the National Center on Health, Physical Activity and Disability (NCHPAD). Further details regarding the DHC project are published elsewhere including the model used for implementation and inclusive PSE outcomes (e.g., increasing accessibility of active community recreation locations, increasing access to adaptive equipment, and implementing inclusive policies) [10].

Over the course of the DHC project, coaches experienced a range of barriers and facilitators that affected their ability to implement PSE changes and their behaviors relating to implementation. Previous research has identified that the behaviors of implementers can affect implementation and that these behaviors can be affected by both internal and external factors [13,14]. To our knowledge, few studies have explored the behaviors of those implementing disability-inclusive PSEs. Understanding the context and behaviors of local health professionals working towards implementation of inclusive PSEs can help to identify what is needed to better support health professionals seeking to improve health and reduce health disparities for PWD. In this study, leading the implementation of inclusive PSEs within the community was the behavior of interest. The purpose of this qualitative study was to better understand the behaviors of community coaches who led the planning and implementation of disability-inclusive PSEs within a community context. Using both the Theoretical Domains Framework (TDF) [13] and the COM-B framework, which identifies Capabilities, Opportunities, and Motivation as factors that affect Behavior [15], we identify factors that impacted implementation of inclusive PSEs.

## 2. Methods

### 2.1. Research Design

This study used a qualitative approach (semi-structured interviews) to examine the barriers and facilitators impacting the coaches during the ongoing process of planning and implementing inclusive PSEs in their respective communities. Interviews were conducted with a purposive sample as part of a process evaluation. The interview guide asked coaches to discuss the implementation process, the barriers and supports that influenced implementation of PSE changes, perceived reach, effectiveness, and sustainability of PSE changes, and experiences working with the community and community organizations. At the time of the interview, coaches had been working through the six-phased implementation model for approximately one year.

### 2.2. Participants

Community coaches, also referred to as implementers, were individuals who led the PSE implementation projects locally in each community. At the start of the project, each community was led by two community coaches. This set of coaches represented a partnership between the local public health department and a local disability-serving

organization. Community coaches (n = 19 coaches) who were active in the DHC project from 2016–2021 participated in the semi-structured interviews. The 19 coaches represented ten communities in 5 US states (Iowa, Montana, New York, Ohio, and Oregon). One interview was conducted per community (n = 10 interviews). In nine communities, this meant that the two coaches representing the community were interviewed together as a team. The tenth community only had one active coach at the time of the interview. Coaches for each community included one from the local public health department and one from a local disability-serving organization apart from one community that only had one coach at the time of the interview.

### 2.3. Data Collection and Analysis

During March and April 2017, semi-structured interviews (n = 10) were conducted with coaches of each community as part of the process evaluation of the funding opportunity. Interviews used video conferencing software and were administered by a research team member not previously involved with the project. To mitigate the potential for bias, the interviewer was external to the primary funding organization and unknown to the study communities. All interviews were audio-recorded and transcribed verbatim.

Using the definitions from the TDF framework, a codebook was created, which guided the content analysis. Transcribed interviews were coded using directed content analysis into the domains of the TDF [13] by two independent researchers using NVivo software (Version 12, QRS International). Through an iterative process, codes were discussed and collated. A third researcher resolved any disagreements for which consensus could not be reached. The codes of TDF domains were then categorized using the COM-B model [15]. These frameworks were chosen to help identify factors that contribute to implementation behaviors. The TDF, which consists of domains including both internal and external factors, has been used to explore implementation across various contexts [16,17]. The TDF can be overlaid with the COM-B model of behavior change [15] (Table 1), which presents broader categories (capabilities, opportunities, and motivations) that contribute to a given behavior. The COM-B model is helpful for organizing the results and our interpretation of the findings. Implementation behavior can be a result of barriers and facilitators across these TDF and COM-B domains [18] and together, these frameworks allow for a better understanding of how to support implementors. Analysis of the interviews was approved by the Institutional Review Board of the University of Alabama at Birmingham.

**Table 1.** Frequency of each code within the domains of the Theoretical Domains Framework (TDF) [13] organized by the COM-B framework (Capabilities, Opportunities, and Motivations that can influence Behavior) [15].

| COM-B Domain | TDF Domain | Frequency |
|---|---|---|
| **Capabilities** | | |
| | Knowledge | 44 |
| | Cognitive and interpersonal skills | 24 |
| | Behavioral regulation | 18 |
| | Memory, attention, and decision processes | 4 |
| | Physical skills | 2 |
| **Opportunities** | | |
| | Environmental context and resources | 185 |
| | Social influences | 136 |
| **Motivations** | | |
| | Optimism | 107 |
| | Beliefs about capabilities | 82 |
| | Social/professional role and identity | 75 |
| | Beliefs about consequences | 49 |
| | Intentions | 47 |
| | Reinforcement | 35 |
| | Goals | 22 |
| | Emotion | 12 |

## 3. Results

Coaches were adults who were employed either by the local public health department or a community-based disability-serving organization and were more often female (n = 13). Table 1 summarizes the frequencies of coaches' references to each TDF domain. Though frequency alone is not sufficient to understanding the community coach's implementation behaviors, frequencies can help identify important domains to further explore. The TDF domains referenced most frequently were 'environmental context and resources' (e.g., funding, time, tools) and 'social influences' (e.g., community awareness, champions, partner networks), which were cited 185 and 136 times, respectively, over the 10 interviews and together represent the opportunity domain of the COM-B. Optimism, within the motivation domain of the COM-B, was mentioned 107 times throughout the 10 interviews. The capabilities domain, comprising 'physical skills' (the physical ability to do something), 'cognitive and interpersonal skills' (other types of non-physical skills), 'knowledge' (awareness of something), 'behavioral regulation' (changing the implementers' actions), and 'memory, attention and decision processes' (retaining and selecting information related to the behavior), was mentioned the fewest times collectively, indicating capabilities may have played a lesser role as either barriers or facilitators to implementation. Results are presented via the COM-B domain in order of importance as determined by the frequency with which the domains were discussed by the coaches and subsequently by the TDF domain. Representative quotes of reported facilitators and barriers to implementation can be found in Tables 2 and 3, respectively.

**Table 2.** Commonly reported facilitators to PSE implementation categorized by domain.

| COM-B Domain | TDF Domain | Emerging Themes | Representative Quotes-Facilitators |
|---|---|---|---|
| Capabilities | | | |
| | Knowledge | Exposure to existing adaptations for inclusion | "…I got to see a lot of things that made working out in a gym a lot easier that were not hard. You know, I never thought about those great big ropes that you shake up and down, well they have them up there! They have them upstairs and I never even thought about taking them downstairs for people in wheelchairs. So it's so nice because I would have never thought of stuff like that…" |
| | | Increased knowledge of inclusion issues | "I think the major thing that I saw and was really a little shocked by is that when we did the CHII assessment after going back and looking at them and viewing them, seeing how negligent—we hadn't done a better job of inclusion in our community" |
| | Behavioral regulation | Identifying ways to include disability | "it really dawned on me that a lot of the other projects I'm currently working on; I could easily look at accessibility when I'm out in the community. And I didn't do that before." |
| | Cognitive and interpersonal skills | Previous experience | "But [coach] had been involved in doing those kinds of assessments before so she really understood it. I think someone who had never seen an assessment like that before would be a little overwhelmed" |

**Table 2.** *Cont.*

| COM-B Domain | TDF Domain | Emerging Themes | Representative Quotes-Facilitators |
|---|---|---|---|
| **Opportunities** | | | |
| | Environmental context and resources | Technical assistance | "the ability to communicate you know, our state expert coach has been amazing. You know, she is readily available for questions at any time I can email her at any time and get a pretty you know immediate response from her but I I've gotten that same-same thing from NACDD and from NCHPAD. They have been willing to answer questions, provide information guidance, so I don't really know that anyone is more valuable." |
| | | Existing community initiatives | "…I was pondering how we were going to lay out our project and then kind of a dawning on us that we had three major community undertakings underway" |
| | Social influences | Existing relationships | "we in our community have great working relationships and strong coalition so I think already having those established relationships because I think every one that we did, we knew somebody from that organization that we could make a phone call to. And that's probably why we're so successful in getting in there. We weren't just making cold calls, we actually knew these people that were calling and we had prior working relationships with them on you know, whatever project" |
| | | Local champions | "…you still get some communities where somebody just has a burning desire, as you do, you know, in your belly for this work. And so it makes it easier if you have a champion…Our champions have been a lot of elected officials." |
| | | Openness to engage in conversation | "I think the CHII assessment process was very useful in making connections with organizations and figuring out, it was basically a good foot in the door to have conversation about what they might be open to doing with our project and you know helped us get a better sense of their readiness to engage with implementation so it seemed to work well as a step I think." |
| | | Willingness to learn | "The biggest thing would be, what's the openness to, you know, see what changes needed to made or what changes could be made to impact others. And so as long as they were open to talking to us, and looking at what information we had, we could do great things" |
| **Motivations** | | | |
| | Reinforcement | Experiencing positive outcomes or feedback | "You just talk about it for a few minutes and it's amazing for people who are going, 'Oh yeah, I should be doing my part'" |
| | Beliefs about capabilities | Perceived self-sufficiency | "But since then, you know, that was at the beginning of the project, we really haven't had a huge need for technical assistance…and process probably gave us a better idea of what-what things we need to look into a little bit more, but we'd already had some tools that we were comfortable using as resources." |

**Table 2.** *Cont.*

| COM-B Domain | TDF Domain | Emerging Themes | Representative Quotes-Facilitators |
|---|---|---|---|
| Motivations | | | |
| | | Realistic goals and expectations | "I think we just set realistic goals. We knew what we could accomplish within our community." |
| | Optimism | Beliefs about PSE change sustainability | "I think that the PSE changes that we are hoping for will be really sustainable. And I think we have the capacity to be able to push for those changes after the grant period is over." |
| | | Belief of overall positive community impact | "I think the changes are going to make the whole community healthier and not just affect people with disabilities that make everybody healthier." |
| | Beliefs about consequences | Positive outcomes for the community | "I think it's prompting a bit of a shift culturally" |
| | | Importance of conversations | "I think the way you approach the assessment overall has a lot to do with it. You know we weren't coming out to do anything punitive, to tell them 'you are going to be fined for this because you are doing something wrong'. It was really more of like, this is educational, we just really want to see what you are doing here and what could be improved in order for you site to be more accessible to our community members who are living with disabilities" |
| | Social/professional role and identity | Previously established roles | "Probably because we both lived here our whole lives… we have served on so many committees and boards and we are present in the community ourselves so it just kind of makes it easy." |
| | | Passion for work | "you know, we—we have, I think, a good amount of passion for the work. And I think that we have that fire in our bellies that we want to continue the work" |
| | Intentions | Intentions to work towards project goals | "…to continue with some of the additional ideas and improvements and suggestions that we are going to be looking for funding for …projects that we can, as a coalition continue to rally around and work towards that common goal" |
| | | Intentions for long term change | "…we saw the state working with the county, the projects kept changing. So our focus was, regardless of what they end up with, how are we going to affect change on projects like this in our community for the long-term…" |
| | Goals | Setting clear goals | "…a couple of simple ones I think we will be able to get done in the not so distant future….there is staff right at [city department] that is kind of looking into the recommendations and doing a cost analysis to see what some of those, sort of quick wins might be" |

**Table 3.** Commonly reported barriers to PSE implementation categorized by domain.

| COM-B Domain | TDF Domain | Emerging Themes | Representative Quotes-Barriers |
|---|---|---|---|
| Capabilities | | | |
| | Cognitive and interpersonal skills | Lack of practice/experience | "I think [the assessment] was a little bit cumbersome at first but once you got used to it, it was pretty easy to use out in the community" |

**Table 3.** *Cont.*

| COM-B Domain | TDF Domain | Emerging Themes | Representative Quotes-Barriers |
|---|---|---|---|
| Opportunities | | | |
| | Environmental context and resources | Lack of time | "...there are a lot of times that I felt like I didn't do as much as I would have liked to, that my time was limited. Even now, I feel like there are so many more things that maybe could have been accomplished if I had more time or more people to help me accomplish those things" |
| | | Usability of resources and tools | "Once I actually got there the whole book then there's a big chart that was printed out for us and it was really hard, kind of hard to navigate and it was on ledger size paper and it came through and that was hard to put all together. So even if that could have been given to us earlier maybe or maybe earlier or something or in an easier format to navigate even maybe." |
| | | Lack of funding and personnel | "I mean we have partners, especially at the city that we have been working with and they have been doing some work as well but it's not like we have other people, we don't have interns or anything. We have kind of done all of the work." "...there was a funding gap ...for those system changes, it required a larger pot of money than what we had originally allocated..." |
| | Social influences | Resistance to change | "This town is tough in regards to people agreeing to go the extra mile and I think [this project] has kind of put feet to the fire." |
| | | Lack of awareness of disability/accessibility | "And so for, you know, a smaller community with numerous small businesses, those businesses are just, you know, they're not educated enough to understand that" |
| | | Fear of cost to address accessibility | "I think that people are scared that if they, you know, somebody comes in and find that maybe there's something that they're not doing correctly...if we find a problem that they're somehow going to be, you know, in some kind of trouble, or there's going to be some major financial requirement that they're going to have to come up with." |
| Motivations | | | |
| | Beliefs about capabilities | Lack of confidence in using recommended tools | "And the most ... time consuming part of the project was just trying to understand that huge document and what it is and the resources in it, and, and how it can be useful to our project partners. Right. So it was it was kind of that in between, you know, after we did the assessments or starting kind of that planning and prioritizing the project and understanding the data that that was the most difficult part," |
| | | Unclear expectations | "I haven't done a lot of work where I have had to make that action plan and have to think through all of those separate steps to get to it and I was honestly starting to get kind of worried like, hey uh oh we haven't done, we haven't followed this exactly." |

### 3.1. Opportunity

The process of implementation was heavily impacted by opportunity. This COM-B domain identified that elements of both the physical and social environment influence implementation behaviors.

#### 3.1.1. Environmental Context and Resources

The most reported influences on implementation of inclusive PSEs were environmental context and resources, meaning that the community context played a role in creating opportunities for implementation. Some communities were able to work with existing implementation projects through which inclusion efforts could have a big impact on the community. In the process of planning the PSE changes, one coach shared about the opportunity to integrate inclusion into existing opportunities, "...I was pondering how we were going to lay out our project and then kind of a dawning on us that we had three major community undertakings underway". Others were able to use resources available in their community to assist in implementation, such as additional help from local organizations (i.e., volunteer organizations, schools, and healthcare systems).

Resources acted as a barrier or facilitator depending on the availability and ease of using the resource. These included experts who provided technical assistance (state experts, national organizations), tools and products that were used to work through implementation, capacity to do the work (time and funding), and human capital contributing to the work. Access to resources served as a facilitator. All communities reported that they were provided tools (e.g., assessment tools, tools for identifying inclusion strategies) and assistance that they needed to work through the implementation process and were able to utilize them whenever they were needed. The most sought-after resource was time. Most coaches indicated there was not enough time to complete everything they wanted. For example, during a discussion of engaging community organizations for implementing inclusive changes, one coach felt that limitations in time and resources limited the number of changes that could be implemented "There are a lot of times that I felt like I didn't do as much as I would have liked to, that my time was limited. Even now, I feel like there are so many more things that maybe could have been accomplished if I had more time or more people to help me accomplish those things". Communities also requested more resources, identifying that additional people to complete the work would have allowed them to complete more activities related to the project. Additionally, funding to cover both the time of those working on the project and fund the PSE changes themselves would have helped facilitate implementation. Some communities were able to secure additional funds to support implementation which helped the project overall.

The usability of some of the suggested templates and products acted as a barrier in some communities. The formatting of some provided resources, such as the tool to help identify inclusion strategies, required additional time and effort to understand and use. For instance, when describing the tool to identify inclusion strategies, one coach said, "...there's a big chart that was printed out for us and it was really hard, kind of hard to navigate and it was on ledger size paper ... that was hard to put all together". Most communities felt the templates and resources themselves were beneficial but could be cumbersome and required assistance at times to use effectively.

#### 3.1.2. Social Influences

Social influences also acted as either a barrier or facilitator depending on the views of the individuals, organizations, or culture within the community. Towns that self-identified as rural or "small town" noted this culture could either challenge or facilitate the implementation process. In some cases, the small-town context contributed to the resistance to change. In other cases, a small town presented more opportunities to leverage personal relationships and professional networks. Regardless of community size, existing relationships also led to increased community buy-in for implementing inclusive PSEs. If there was already a positive working relationship with community members and organizations, the coaches

found that their community was more likely to support changes. One coach acknowledged the positive effect previous relationships had in building support for inclusive PSE changes and working with various organizations saying, "we in our community have great working relationships and strong coalitions so I think already having those established relationships...we knew somebody from that organization that we could make a phone call to. And that's probably why we're so successful in getting in there. We weren't just making cold calls, we actually knew these people that were calling, and we had prior working relationships with them on you know, whatever project". Examples of existing relationships that facilitated implementation included local leadership, elected officials, and a variety of local service providers. Additionally, identification of champions or "movers and shakers" that would promote disability inclusion facilitated progress toward implementation.

Alternatively, some communities reported that influential individuals or groups negatively impacted progress toward implementation and were resistant to change. A lack of awareness about disability inclusion could also lead to fear, primarily the fear that learning more information would require implementing expensive changes. One coach described the fear saying, "I think that people are scared that if they, you know, somebody comes in and find that maybe there's something that they're not doing correctly, ...you know, if we find a problem that they are somehow going to be, you know, in some kind of trouble or there's going to be some major financial requirement that they're going to have to come up with". Some coaches were able to overcome this obstacle through deeper conversation, relationship building, and training. These efforts allowed community coaches to gauge the interest of organizations who were initially hesitant in future inclusive implementation projects. In some communities, the lack of awareness was met with a willingness and desire to know more about how to address the issues related to inclusion. This facilitated creative solutions and capacity building within these communities through fostering relationships between individuals, organizations, and disability and health experts.

### 3.2. Motivation

#### 3.2.1. Optimism

Optimism was reported as a facilitator of the implementation process. The belief that there will be an overall positive impact of inclusive PSEs encouraged continued steps towards implementation. This optimism was not limited to the impact of the PSE change itself, but also the positive ripple effect that the process of implementing inclusive PSEs would have throughout the community. Regarding PSE sustainability, one coach noted "I think that the PSE changes that we are hoping for will be really sustainable. And I think that we have the capacity to be able to push for those changes after the grant period is over". Belief that the inclusive PSEs would be sustainable acted as a facilitator and motivated community coaches to continue through the process.

#### 3.2.2. Beliefs about Capabilities

Beliefs about their ability to implement inclusive PSEs and work through the steps associated with implementation acted as both a barrier and facilitator for the coaches. When coaches perceived self-sufficiency, it often paired with positive beliefs about their ability to implement PSEs. They believed that with the skill set they already possessed, they could continue to work through the steps toward implementation. When the coaches felt that implementation aligned with their skillset and their context, coaches felt confident in their abilities to implement inclusive PSEs. However, lacking confidence in using the recommended tools and unclear expectations of the work to be done as part of the project acted as barriers to implementation. Some communities viewed the ability to reach out to technical assistance as a great asset to implementation. It allowed the community coaches to feel more comfortable in their capabilities, knowing that there was support if it was needed.

### 3.2.3. Professional and Social Role and Identity

When coaches identified that this project's work aligned with their professional duties or with their typical role within their community, it facilitated implementation. This included established roles through previous work in their professional capacity or service to the community. In one community, the coaches acknowledged the positive impact of their previously established roles within the community which developed over time and through being active within the community, saying "probably because we both lived here our whole lives…we have served on so many committees and boards, and we are present in the community ourselves so it just kind of makes it easy". This sentiment and presence within the community also aligns with the importance of existing relationships. Some also identified a passion for disability inclusion work that aligned with their work in public health.

### 3.2.4. Goals and Intentions

Having both inclusion goals and intentions promoted inclusive PSE implementation. Coaches stated intentions related to identifying groups to engage in the implementation process or intent to implement inclusive changes both as part of this project and in future work. Goals were more specific plans of action that the coaches were pursuing regarding implementation. Some coaches identified that this clear picture of what to implement was beneficial in feeling successful. It should be noted that part of the 6-phase model was to develop a community action plan which included writing specific and measurable PSE goals. Communities often pointed to specific examples within their community action plans or where steps towards successful implementation of inclusive PSEs were occurring.

### 3.2.5. Beliefs about Consequences

Beliefs about consequences mostly related implementation processes to positive outcomes. Many believed that the work that they were doing towards implementation would lead to positive changes in the community and for the health of PWDs. They also believed that working towards implementation would lead to positive cultural changes and conversations about access and inclusion. Coaches noted that these conversations should be approached in an unassuming and educational manner, but the opportunity for these conversations was an overall positive consequence of implementation.

Coaches also identified that the progression through phases of the model was important to implementation. This included a commitment phase, an assessment and training phase, a prioritization and planning phase, and including PWDs throughout. The inclusion of PWDs in the processes was believed to positively impact inclusive implementation.

### 3.2.6. Reinforcement

Coaches identified that experiencing or observing positive outcomes of actions related to implementation encouraged continued action. For example, positive responses to conversations about disability health or sharing success stories within communities helped to reassure coaches that they were having an impact within the community.

### 3.3. Capabilities
### 3.3.1. Knowledge

Generally, coaches identified that the implementation process increased their awareness of where there were inclusion issues within their community. As they became more aware of areas their communities could improve and potential solutions, they felt more capable of implementing PSEs. Acknowledging that there was more to learn was deemed "eye-opening" by many coaches and was typically related to assessment results. The assessment also highlighted good inclusive practices that were occurring within the community.

One training was held at an inclusive physical activity facility where all the coaches came together and learned about the implementation processes. Coaches cited this exposure to inclusive physical activity as a facilitator as it showed examples of inclusion solutions

that they might be able to bring back to their communities. For instance, one coach was able to recognize some simpler changes that could be implemented within a local fitness facility, saying, "...I got to see a lot of things that made working out in a gym a lot easier that were not hard. You know, I never thought about those great big ropes that you shake up and down . . . they have them upstairs and I never even thought about taking them downstairs for people in wheelchairs". Exposure to a location with new and different solutions helped to expand the coach's knowledge of what could be implemented.

### 3.3.2. Cognitive and Interpersonal Skills

Previous assessment and implementation experience facilitated the implementation process. Where previous experience was lacking, coaches identified that practice was necessary along with additional training to help to understand the steps of the process and feel confident in performing the necessary tasks. This was typically related to conducting community assessments and prioritizing solutions identified by the assessment data.

### 3.3.3. Behavioral Regulation

Many coaches identified that they approached their work differently after becoming more aware of the needs of PWDs and how to include disability. One coach noted, "It really dawned on me that a lot of the other projects that I'm currently working on; I could easily look at accessibility when I'm out in the community. And I didn't do that before". Coaches discussed restructuring how they approach their work to be more intentional about inclusion.

## 4. Discussion

This study sought to understand barriers and facilitators that impacted the behaviors of those implementing inclusive PSEs. Factors within environmental context and resources followed by social influences were the most frequently referenced TDF domains within the 10 interviews, indicating that availability of opportunities and resources facilitated inclusive PSE implementation. The opportunities available and the social context of each community were identified by all coaches in both positive and negative lights, indicating a need to further explore the opportunities related to inclusive PSE implementation.

Lack of awareness regarding disability within the broader community was a prevalent barrier in many communities. Previous research has identified that the lack of information on disability inclusion among community members prevents PWDs from engaging in physical activity within the community [7,19]. Specifically, community members and other professionals implementing community changes need to better understand the 'why' behind inclusion and accessibility requirements [7,19]. To facilitate better opportunities for inclusion, training, and incentivization are recommended interventions that can be implemented at the community level [15]. Additionally, it is valuable for those with expertise in disability inclusion to be meaningfully engaged when a proposed community or public health change is planned. Identifying inclusive practices from the beginning can save resources in the long run [10] and more efforts to provide training and raise awareness may encourage disability inclusion.

Professional networks and relationships played a large part in creating a social environment that supported assessing accessibility and inclusion and implementing inclusive PSEs into new or existing programs/initiatives. In communities where the coaches had previous involvement with the organizations implementing changes and maintained a positive relationship, it was easier to accomplish the goals of the project. In communities where those relationships did not yet exist, relationship building impacted the timeframe for implementation. A community's capacity built from previous experience implementing PSEs and relationships with key stakeholders was critical to implementation. Similarly, one case study which explored the resources necessary to implement PSEs found that including stakeholders with appropriate expertise as well as dedicated funding to support implementation were essential components of a program's capacity for PSE implementa-

tion [20]. Another study noted the importance of including community leadership in PSE changes [21]. It is vital to recognize the significance of social influences in inclusive PSE implementation.

Coaches' motivations were also important in driving implementation behavior, specifically, optimism towards implementing beliefs that the end goal of disability inclusion would have positive outcomes within the community and alignment with professional roles and capabilities. This may be related to the phased implementation approach and the significant training component coaches participated in which encouraged understanding the importance of inclusion and its impact. A previous study has shown that using a structured and supported implementation process (including technical assistance, training, and a theory-driven process) promoted successful PSE implementation over time compared to a less structured approach [22]. The grantees in the DHC project followed a similar structure which likely contributed to both capability and motivation to implement PSEs.

Technical assistance also played a role in supporting these capabilities and motivations and reinforced that experts should be made available to those trying to implement inclusive changes in their community. Technical assistance is valuable in sharing skills and identifying best practices for PSE implementation [12]. It often helps to have outside individuals offering additional perspectives in the decision processes and this can encourage coaches as they work through implementing larger changes.

### 4.1. Implications

To understand how to support implementors, both the COM-B and the TDF can then be overlaid with the behavior change wheel (BCW) [15], which aligns each domain with specific, behavioral interventions. In this study, factors in the opportunities domain influenced implementation behaviors for which the BCW suggests interventions such as incentivization and persuasion. Incentivization calls for a reward or prize structure [15]. At the community level, incentivization could involve offering grants or other funding mechanisms to organizations for implementing inclusive PSEs. Though some disability-specific funding mechanisms exist to support community inclusion, funding amounts are often limited which in turn limits implementation of larger PSE changes which may require funding for successful implementation. Therefore, maintaining consistent streams of funding for inclusive PSEs is important to incentivize disability inclusion work within communities.

Persuasion involves leveraging communication to change feelings towards implementation or stimulate action [15]. This may be suited to support implementors within communities lacking awareness of the needs of PWDs. Providing resources to help implementors focus their messaging could promote buy-in for inclusive PSE changes within the broader community and within specific community organizations.

### 4.2. Limitations

Interviews were conducted primarily with pairs of coaches which facilitated complete answers as coaches reminded each other of processes and examples but limited our ability to understand the relationship between coaches and if that impacted implementation. Additionally, there was a potential for response bias as the interviews were conducted in conjunction with an evaluation by the funding agency. However, the interviewer did not have any prior contact with the communities and was not affiliated with the primary funding organization to try to mitigate this potential for bias.

## 5. Conclusions

This identifies barriers and facilitators implementors experienced while implementing disability-inclusive PSEs. Factors within the opportunities domain were most often reported to influence the community coaches' ability to implement inclusive PSEs. Providing support including funding, staffing, and other resources can help to facilitate opportunities. The social climate of a community must also be considered when planning and implementing

PSEs inclusive of disability. This might involve raising awareness of the needs of PWDs and strategically engaging partners both in disability-serving organizations and in positions of influence and power. Future research should explore strategies that increase opportunities for disability-inclusive PSE implementation. Identifying ways to facilitate inclusion within PSEs allows individuals with disabilities to participate within their communities and can improve overall health outcomes for PWDs through increased access to all health promotion opportunities.

**Author Contributions:** Conceptualization, C.H., K.V., K.E. and Y.E.; Methodology, C.H. and K.V.; Formal Analysis, C.H. and K.V.; Writing—Original Draft Preparation, C.H. and K.V.; Writing—Review and Editing, C.H., K.V., K.E. and Y.E.; Supervision, K.V. and Y.E.; Project Administration, C.H., K.V., K.E. and Y.E.; Funding Acquisition, K.E. and Y.E. All authors have read and agreed to the published version of the manuscript.

**Funding:** Funding for this project was provided to NACDD through the National Center on Birth Defects and Developmental Disabilities, Centers for Disease Control Grant #U38OT000225 with additional support from the National Center on Health, Physical Activity and Disability (NCHPAD), Cooperative Agreement Number, 5NU27DD001157, funded by the Centers for Disease Control and Prevention. The post-doctoral research associate was also supported by the National Institute on Disability, Independent Living, and Rehabilitation Research, Administration for Community Living, (grant #90ARCP0004), Advanced Training in Translational and Community Engaged Scholarship to Improve Community Living and Participation of People with Disabilities, University of Illinois Chicago. The findings and conclusions in this report are those of the authors and do not necessarily represent the official position of the National Center on Birth Defects and Developmental Disabilities, Centers for Disease Control and Prevention and do not necessarily represent the policy of the U.S. federal government.

**Institutional Review Board Statement:** This research was approved by the Institutional Review Board of the University of Alabama at Birmingham (protocol 300001985). The interviews were collected in a non-research capacity as part of a process evaluation. Ethical approval was received prior to the analysis of the interviews, which was the research portion of this project.

**Informed Consent Statement:** Informed consent was waived as the project was determined to be exempt by the Institutional Review Board at the University of Alabama at Birmingham and data did not include identifiable information.

**Data Availability Statement:** The data presented in this study are available on request from the corresponding author.

**Acknowledgments:** The authors are grateful to the National Association of Chronic Disease Directors and the community coaches involved in this project who shared their experiences as they promoted disability inclusion within public health. We thank the National Center on Health, Physical Activity and Disability staff for their valuable technical assistance and sharing their knowledge and expertise in disability inclusion. We would also like to acknowledge Dori Pekmezi, who provided critical support for this manuscript.

**Conflicts of Interest:** The authors declare no conflicts of interest.

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
