# Peer review of "Perceived Barriers and Facilitators to Implementation of Inclusive Policy, Systems, and Environmental Changes"

_disabilities, doi:10.3390/disabilities4010003_

Round 1

Reviewer 1 Report

Comments and Suggestions for Authors

This paper provides a qualitative analysis of in-depth interviews with people who work in public health and disability-serving organizations as they describe the barriers and facilitators to carrying out inclusive policy, systems, and environmental changes for health-promoting activities in their local communities. This topic is of great practical relevance – whether to similar programs or to programs that do not have a disability focus but who wish to identify methods that ensure inclusion of people with disabilities.

This paper does a good job of setting up the importance/relevance of the topic in an introductory summary without belaboring the details. Overall, I find it worthy of publication. I do have some suggestions to offer clarity to the reader and to ensure it has the most utility.

First, I think the article would benefit from some more background on the DHC program, or at least, on the role of the coaches. I find the term “coach” or “community coach” to not be very descriptive in this context.  Did the coach participate in identifying relevant PSEs, did they carry the administrative load of carrying them out or the practical workload of interacting with the public? I think this would best fit in around line 50 where some detail on the coaches is introduced (there is a bit more around line 107, but it feels misplaced here). Ideally it would be useful to see what types of public health and disability-serving organizations coaches worked, or any other background you have about them (gender is mentioned later), their experience, or type of work they contributed to this program.  Later I see the term “implementers” on page 2 (line 59). Can we think of coaches as implementers? As a reader, I’m not sure because I don’t quite know what the coach’s role is. This context is important given the limited background to the program they represent.

Beyond this, the methods are generally well described and thorough, although I would like some more background on the interview itself. Specifically, what questions were asked? The TDF and COM-B frameworks seem uniquely appropriate for this analysis. One observation: “COM-B” was not defined until Table 1 it would be useful to spell it out first use. While I found Table 1 to be helpful to familiarize myself with these COM-B domains, it is a bit redundant with Table 2. I would suggest removing table 1 and using table 2 to both introduce the domains and summarize the data.  If appropriate to the authors, consider reordering Table 2 to most prominent TDF domain within COM-B domain for a more logical read.

On line 123, the authors describe how the results are presented by domain. It would be helpful to specify “presented by domain in order of frequency” or “in order of importance,” since they are not presented in the “COM” order. Table 3 displays the qualitative data nicely. I would find the summary of the results to be easier to interpret if it directly referenced table 3 or table 2 – that is, how many times was “environmental context” referenced by coaches (or by how many coaches)? More specifically, what are some examples of this. As a reader, it is not easy to move from the text summary to Table 3. This is in part because it is qualitative and not quantitative, but also because it is not ordered the same as in text (neither by domain nor subdomain). Providing specific quotes/excerpts in text would be very beneficial.

I found that the discussion section went into more detail than the findings summarized. In particular, lines 288-294 reveal patterns not described in the analysis. I think this could be rectified by incorporating suggestions above.

Style/editorial suggestions:

·         Title: Lowercase “And”

·         The acronym for COM-B is not described until Table 1.

·         Line 31 (p1) there is an extra space or two between sentences.

·         Line 52 (p2) has a sentence fragment or else needs to capitalize the start of the sentence.

·         Line 152 “Benefitted” should be benefited?

Reviewer 2 Report

Comments and Suggestions for Authors

It will be recommended that the introduction presents in a clearer manner how this research contributes to the literature and also what evidence exists in the topic.

In the methodology, can you please describe better how participants were selected and why those communities. It is unclear if you had 19 participants, why only 10 interviews, did you conduct group interviews? Did you stop the interviews when you reached saturation? Did you use snowball sampling? Or what time of sampling was used?

In the methodology, you should include how you analysed the data, created categories, transcribed the interviews, and coded them?

Do you have an ethical approval?

The methodology section needs to provide information about the participants, the methods and the analysis. The current version is not complete.

Why to use content analysis and not thematic analysis to analyse the results? If you want to analyse the perceptions of individuals, thematic analysis is a better method.

The results section will benefit from including quotes illustrating the argument in the text. You can use some of the ones in Tables 3 and 4.

Comments on the Quality of English Language

The authors need to improve the methodology and the presentation of the results.

Round 2

Reviewer 1 Report

Comments and Suggestions for Authors

The authors responded to all comments in initial review.

Author Response

Thank you for your additional review of the revised manuscript. 

Reviewer 2 Report

Comments and Suggestions for Authors

It will be recommended to mention in the abstract the city in which this research was conducted.

Across the paper, there is no clear explanation of the analysed programme and the context.

The results section improved a lot with the inclusion of the quotes and tables. However, it will be helpful if the authors use the information in the text and present the potential contradictions and agreements the authors found.

The discussion should improve and discuss the results and the implications of the results.

Given that coaches come from different sectors, it is important to present differences or similarities using the geographic location of the coaches.  

Comments on the Quality of English Language

The language can be improved
